# Cell death and iron deposition in the liver in two murine models of acute radiation syndrome

Dmitry T. Bradfield[1], John E. Slaven[1], W. Bradley Rittase[1], Milan Rusnak[1,2],
Aviva J. Symes[1], Grace V. Brehm[1], Jeannie M. Muir[3], Sang-Ho Lee[4],
Joseph A. Anderson[5], Regina M. Day[1]*

1 Department of Pharmacology and Molecular Therapeutics, Uniformed Services University of the Health Sciences, Bethesda, Maryland, United States of America, 2 Department of Gynecologic Surgery and Obstetrics, Uniformed Services University of the Health Sciences, Bethesda, Maryland, United States of America, 3 Department of Pathology, Uniformed Services University of the Health Sciences, Bethesda, Maryland, United States of America, 4 Pathology Department, Research Services, Naval Medical Research Center, Silver Spring, Maryland, United States of America, 5 Comparative Pathology Division, Department of Laboratory Animal Resources, Uniformed Services University of the Health Sciences, Bethesda, Maryland, United States of America.

* regina.day@usuhs.edu

## Abstract

Different tissues exhibit differential sensitivity to ionizing radiation exposure and display different time courses of pathologies that are not well understood. Ionizing radiation causes hemolysis of red blood cells, causing the release of iron that is taken up by a variety of tissues. The increased iron has been associated with altered expression of iron binding proteins and, in some cases, markers of ferroptosis. Here we examined the time course of iron uptake in murine liver following $^{60}$Co total body irradiation (TBI) at 7.9 Gy ($LD_{90/30}$) and 6.85 Gy ($LD_{0/30}$). 7.9 Gy induced hydropic degeneration, micro-vesicular steatosis, and inflammatory cell infiltration, whereas at 6.85 Gy the livers displayed only inflammatory cell infiltration. In both cases, iron levels increased significantly, maximal at ~21 days post-TBI. Increased iron was associated with altered expression of ferritin, heme oxygenase, an enzyme required for iron recycling, and the pro-inflammatory cytokine serum amyloid A, maximal ~16–21 days. 7.9 Gy induced liver caspase-3 activation consistent with apoptosis. In contrast, 6.85 Gy induced markers of ferroptosis but not of apoptosis. Our data indicate that iron is deposited in the liver at a delayed time point following radiation and is associated with increased ferritin, HO-1, and inflammatory cytokine production.

## Introduction

High-dose total body irradiation (TBI) induces multi-organ damage, collectively known as acute radiation syndrome (ARS) [1,2]. The hematopoietic system is uniquely sensitive to radiation exposure, and hematopoietic subsyndrome of ARS (H-ARS) is characterized by anemia and panleukopenia, leading to tissue hypoxia, immune

**Data availability statement:** All relevant data are within the manuscript and its Supporting Information files.

**Funding:** Defense Medical Research and Materiel Command, Radiation Health Effects Research Program, Joint Program Committee 7, grant numbers DM178018 and 1 I80 VP000264-01 The funders of this study had no role in the study design, data collection and analysis, decision to publish, or preparation of the manuscript.

**Competing interests:** The authors have declared no competing interests exist.

suppression and vulnerability to opportunistic infection, acute inflammatory response, and coagulation dysfunction [1,3–8]. In response to high dose radiation exposure, mature white blood cells and hematopoietic progenitors undergo apoptosis [5,9]. However, red blood cells (RBCs) and reticulocytes lack DNA and the machinery required for apoptosis; instead these cell types undergo hemolysis in response to ionizing radiation, that is believed to be induced by oxidative stress, membrane damage, and protein denaturation [5,10–12]. The destruction of RBCs can contribute to tissue hypoxia as well as the release of potentially toxic iron ions [12,13].

Iron is nutritionally essential and is required for normal cellular function due to its involvement in multiple enzymatic and metabolic activities. However, because of the potential toxicity of free iron, a variety of redundant, highly conserved systemic and cellular mechanisms are in place to maintain iron in protein-bound states and to prevent iron overload in cells and tissues [14]. Iron handling is regulated by multiple mechanisms that transfer iron via transporters, vesicular trafficking, and chaperone proteins. Tight regulation of ferric iron is orchestrated via the transferrin (TF) glycoprotein, synthesized in hepatocytes to transport iron. Most excess iron in the body is stored in hepatocytes, in complex with ferritin [14]. Excessive iron accumulation in cells or tissues can induce ferroptosis in part from the generation of free radicals. Dysregulation of iron has been shown to contribute to cell loss and tissue dysfunction in tumorigenesis, metabolic diseases, blood disorders, and some degenerative diseases [15–18].

The liver plays a key role in the regulation of iron absorption in the intestine and its release from cellular storage, including from specialized macrophages [19]. Under homeostatic conditions, iron is replenished through diet and is absorbed by epithelial cells in the duodenum and upper ileum [20,21]. Regulation of iron on a systemic level requires coordination of several key regulators, most notably being hepcidin, a peptide hormone mainly expressed in hepatocytes [22,23]. Hepcidin expression is regulated by iron and iron-sensing mechanisms and its release from the liver is contingent upon iron abundance in plasma [20,24]. Hepcidin post-translationally regulates ferroportin (SLC40A1), an iron transporter required for iron export from cells from other tissues [20,24]. Additionally, the liver serves as an iron storage depot, in part due to its requirement for large amounts of iron for detoxification enzyme function [19,25]. Hepatocytes take up transferrin-bound iron, and excess iron (iron not required for cellular functions) is stored within a ferritin core [26]. Within the liver, the resident macrophage population, Kupffer Cells (KC) coordinate hepcidin expression and can participate in iron recycling through erythrophagocytosis of damaged RBCs [27]; in this later process, KC recycle the heme iron with the upregulation of heme oxygenase (HO-1). KC have limited tolerance to iron, and can also display iron toxicity [21]. If the normal storage potential of hepatocytes is exceeded, and iron-induced hepatocellular necrosis (sideronecrosis) occurs, KC can become iron laden by phagocytosing necrotic hepatocytes [26]. Over an extended period of time, elevated iron can cause hepatic damage leading to hepatic fibrosis, cirrhosis, and potentially hepatic carcinoma [26].

The C57BL/6 murine model is a widely utilized model of ARS [1,2,28]. We previously demonstrated that radiation-induced RBC hemolysis is associated with increased hemosiderin in the bone marrow and spleen in C57BL/6 mice [1,2]. Iron levels in the bone marrow and spleen were maximal at ~7–14 days post-irradiation [1,2]. Increased hemosiderin in these tissues occurred in the presence of increased levels of hepcidin in the liver following TBI, which would suppress iron export from tissues via ferroportin [1,2]. Here we provide evidence of delayed iron accumulation in the liver in the C57BL/6 murine model of total body irradiation. We utilized two doses of radiation, 7.90 Gy, (90% lethality at 30 day [$LD_{90/30}$]) and 6.85 Gy (0% lethality at 30 days [$LD_{0/30}$]). Although the two doses of radiation resulted in different histological effects, iron deposition occurred in both. Further investigation into transcriptional and translational regulation of iron storage proteins, iron transport proteins, and iron-regulated apoptosis (ferroptosis) revealed differences between the two doses. The data provide evidence for iron release as a potential secondary toxic event involving iron following both lethal and sublethal TBI exposures.

## Results

### 7.9 Gy TBI induces tissue injury and iron deposition in the liver

We previously showed that 7.9 Gy total body irradiation (TBI) results in 80–90% lethality in C57BL/6 female mice [28]. We investigated the time course of radiation-induced tissue alterations in the liver tissue following 7.9 Gy TBI. H&E stained liver sections displayed marked hydropic degeneration of cells at 5 days post-irradiation (Fig. 1A). Tissue alterations at this time point were characterized by large spaces between cells and overall cellular swelling, likely due to osmotic pressure, causing cells to be over-encumbered by intracellular water and hyperplasia. At 12–16 days post-irradiation,

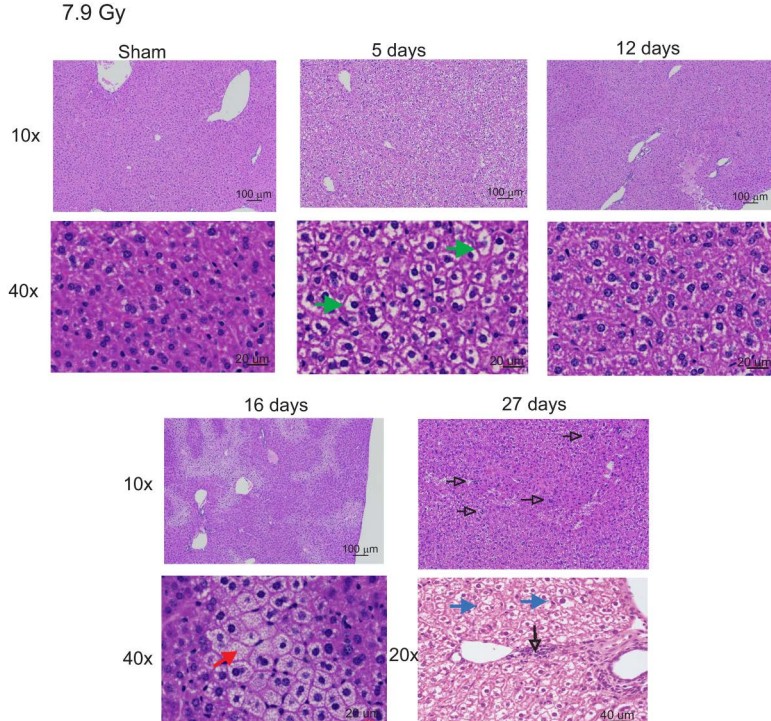

**Fig 1. Histology and iron concentration in mouse livers following 7.9 Gy TBI.** C57BL/6 mice were exposed to 7.9 Gy TBI. At the indicated time points, mice were euthanized, and liver tissue was obtained for histology. Liver sections were stained with H&E to visualize cellular effects of radiation. Representative images from 3 individual animals are shown, with 10×, 20×, and 40×magnification indicated. Green arrows: hydropic degeneration. Open black arrows: macrophage infiltration. Blue arrows: carbohydrate accumulation. Closed red arrow: micro-vesicular steatosis.

hydropic alterations were resolved although minimal to mild hyperplasia remained. At these time points there was significant hepatic micro-vesicular steatosis, usually an indication of increased fat content of liver cells or alterations in metabolic processes. At 27 days post-irradiation, there was minimal, multifocal areas of infiltrated neutrophils and mononuclear cells; these were mostly random, but more often occurred in the periportal regions. Carbohydrate accumulation was also observed in some hepatocytes at this time point, causing reduced cytoplasmic staining.

Prussian Blue staining for iron deposition showed diffuse grey-blue staining, in both intracellular and extracellular regions of the tissue at both 16 and 27 days post-irradiation (Fig. 2A). An iron assay performed on liver tissue showed a steady increase in iron levels in liver tissue that was significant (~2.8-fold, $p < 0.05$) at 21 days post-irradiation (Fig. 2B).

## 7.9 Gy total body irradiation results in upregulation of iron-binding proteins in the liver

We examined changes in expression of iron binding proteins in liver following exposure to 7.9 Gy TBI. Ferritin, the primary storage protein for iron within the liver, showed ~3.5-fold at 16–27 days post-irradiation ($p < 0.05$ for both time points, Fig. 3A), corresponding with the time of increased iron levels within the liver. However, ferroportin, the primary protein for iron export, was not significantly altered over the time course of the experiment (Fig. 3B).

We examined the gene expression of several additional proteins shown to be involved in iron binding and transport. We observed several patterns of gene regulation: 1) biphasic regulation, with an early increase in expression followed by suppression; 2) gene suppression; and 3) no significant change. In the first group with biphasic regulation, transferrin (*Trf*), the primary protein for transport of iron in the serum, displayed ~4-fold increase at 5 days post-irradiation ($p < 0.005$) that returned to below baseline levels by 27 days (Fig. 3C). The feline leukemia virus subgroup C receptor 1a (*Flvcr1*, heme transport [29]) displayed a trend toward increased expression at 5 days post-irradiation followed by reduced expression at days 16 and 27 that was significant compared with day 5 expression ($p < 0.05$, Fig. 3D). The expressions of the *Flvcr2* isoform (also heme transport) and integrin M (*Itgam*, iron oxide particle transport [30]) displayed trends of similar biphasic regulation, but these did not reach significance (Fig. 3E, F).

In the group of genes that were suppressed, the transferrin receptor (*Tfrc1*) showed ~3-fold reduced expression at 16 days post-irradiation ($p < 0.05$, Fig. 3G). Transferrin receptor 2 (*Tfrc2*), a liver specific receptor for transferrin, showed ~3-fold decrease at 27 days post-irradiation ($p < 0.05$; Fig. 3H). The heme carrier protein (*Hcp1*, also called proton-coupled folate transporter, *Pcft*) was reduced by 2-fold at 12 days post-irradiation ($p < 0.05$; Fig. 3I).

Finally, in the last group of genes, Lipocalin 2 (*Lcn2*, alternative for transferrin iron import [31]) and the divalent metal transporter 1 (*Dmt1*, also called solute carrier family 11 member A2 [*Slc11a2*], transport of transferrin-bound and non-transferrin-bound iron [32]) were not significantly changed in the liver following TBI (Fig. 3J, K).

Kupffer cells in the liver can perform erythrophagocytosis and heme recycling. An increase in heme levels induces the upregulation of heme oxygenase-1 (HO-1), a key enzyme for heme recycling [33]. *Ho-1* gene expression was steadily increased post-irradiation, reaching ~4-fold at 16 and 27 days post-irradiation ($p < 0.05$ and $0.01$, respectively) (Fig. 3L). In agreement with the increased gene expression, we observed a steady increase in HO-1 protein levels to 1.5-fold that was significant at 27 days post-irradiation ($p < 0.05$, Fig. 3M). Together these data indicate that there is a delayed increased storage of iron in ferritin in the liver following 7.9 Gy TBI. The upregulation of HO-1 could indicate that recycling of heme may be a primary source of increased iron post-irradiation in the liver.

## 7.9 Gy TBI induces early apoptosis but does not activate markers of ferroptosis

Studies from our laboratory and others suggest that iron released post-irradiation results in upregulation of markers of either apoptosis or ferroptosis [2,34,35]. Western blotting showed that activated caspase-3 was increased ~6-fold at 5 days post-irradiation, but not at later time points (Fig. 4A, $p < 0.05$). Activated caspase-3 was also observed by immuno-histochemistry in liver sections (Fig. 4B). Pronounced staining was observed in the livers at 5 days post-irradiation, while diffuse staining was visible at 12 and 16 days post-irradiation.

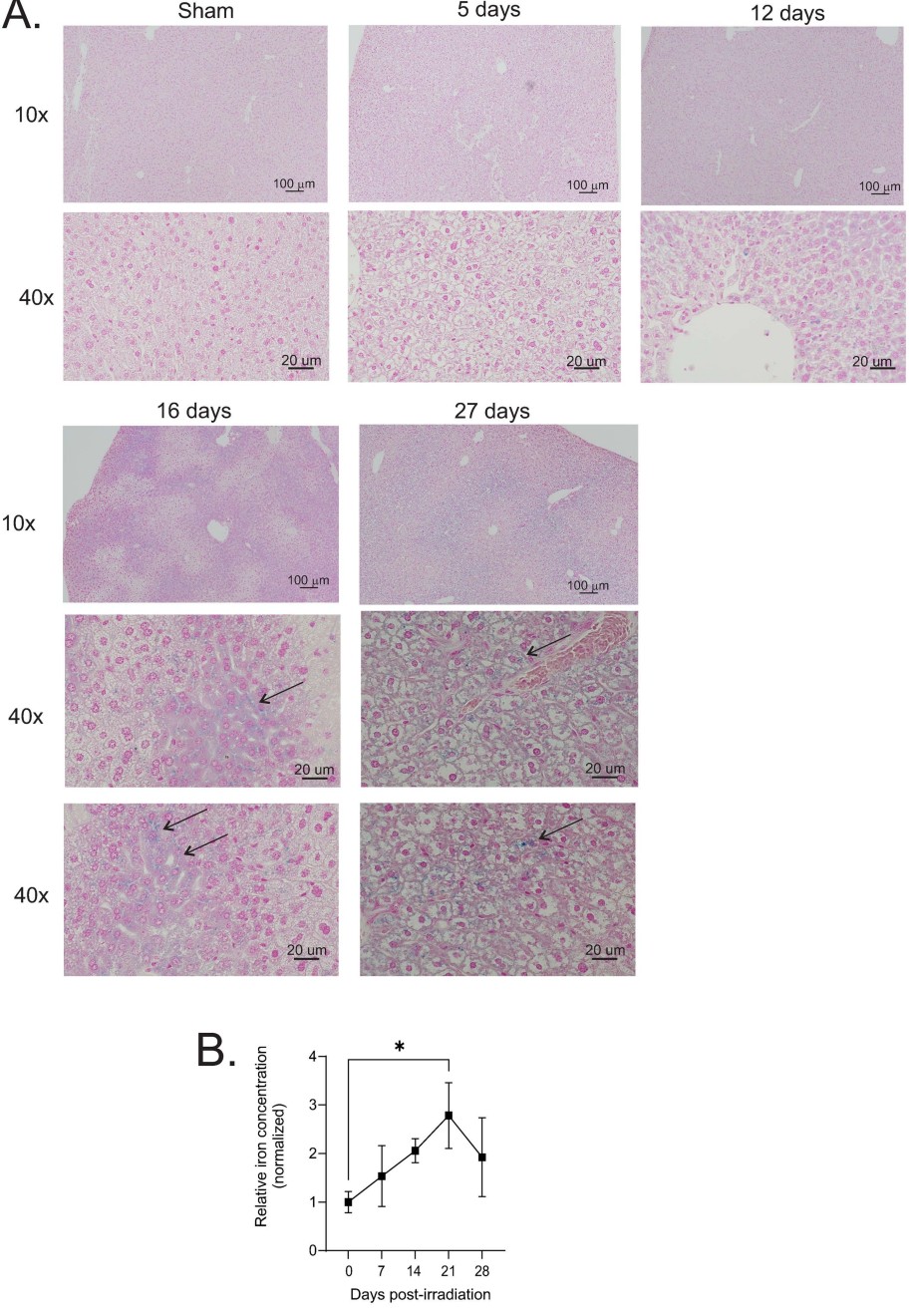

**Fig 2. Iron concentration in mouse livers following 7.9 Gy TBI.** C57BL/6 mice were exposed to 7.9 Gy TBI. At the indicated time points, mice were euthanized, and liver tissue was obtained for histology. **(A)** Prussian blue staining was performed to visualize $Fe^{3+}$ deposition in the tissue. Representative images from 3 individual animals are shown, with 10× and 40× magnification indicated. Arrows indicate regions of iron deposition at 16 and 27 days. **(B)** Tissue iron concentration, normalized to protein concentration. Graph shows means ± SEM for n = 4 animals per group; * indicates $p < 0.05$.

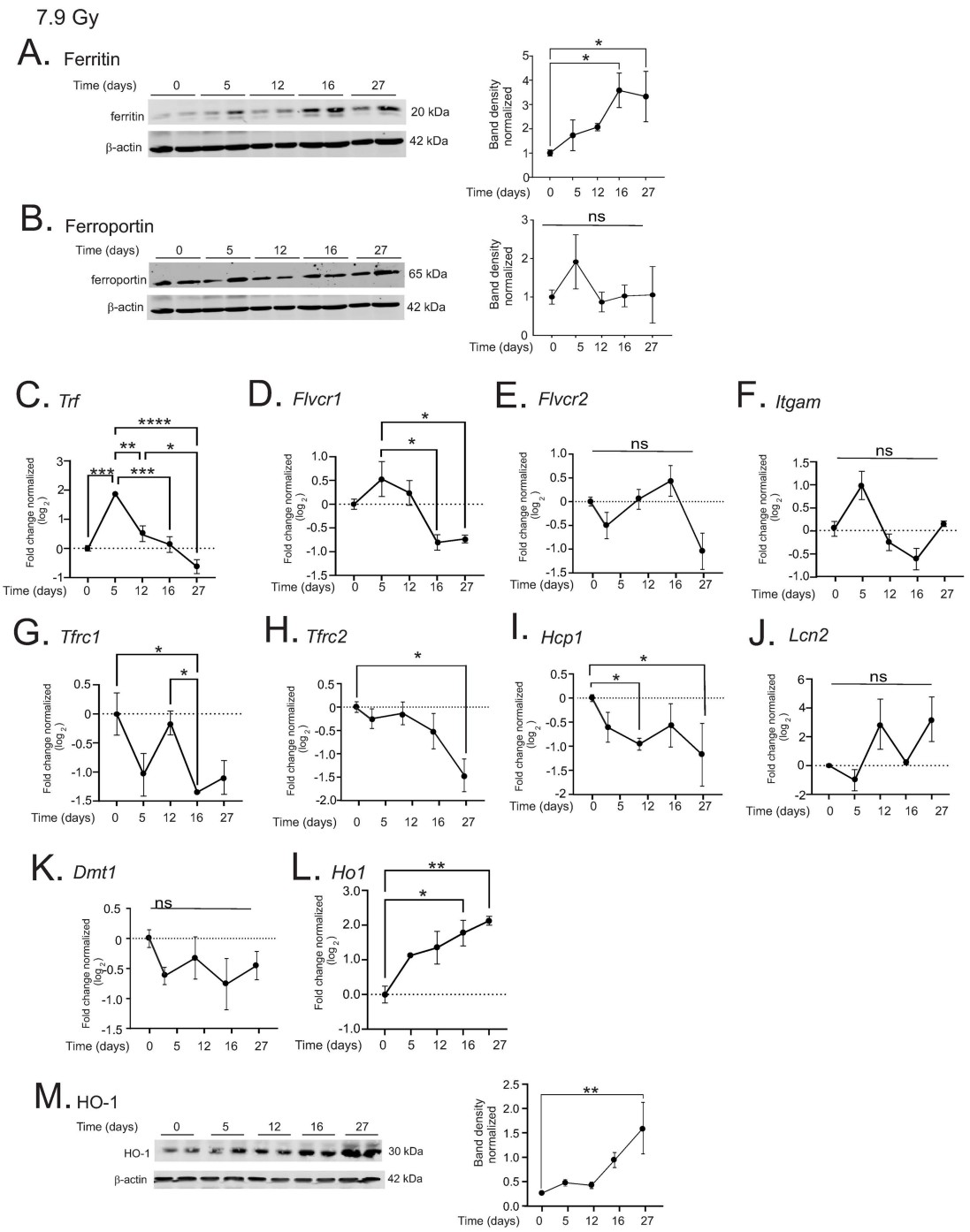

**Fig 3. 7.9 Gy TBI results in increased iron binding proteins in the liver** C57BL/6 mice were exposed to 7.9 Gy TBI. At the indicated time points, mice were euthanized and liver tissue was obtained for protein and RNA analysis. A-B. Western blotting was performed to detect proteins: ferritin (A) and ferroportin (B). Representative blots are shown for n = 4. Graphs show means of band densities ± SEM for n = 4 animals per group, normalized to β-actin protein. C-L. qPCR was performed to determine expression of the following genes: *Trf* (transferrin) (C); *Flvcr1* (feline leukemia virus receptor) (D); *Flvcr2* (E); *Itgam* (integrin alphaM/Mac-1) (F); *Tfrc1* (CD71/transferrin receptor1) (G); *Tfrc2* (H); *Hcp1* (heme carrier protein 1) (I); *Lcn2* (lipocalin-2) (J); *Dmt1* (divalent metal transporter 1) (K); and heme oxidase-1 (*Ho1*; L). M. Western blot for HO-1 protein. A representative blot is shown for n = 4 individual animals. Graph shows band densities normalized to β-actin protein. All graphs show means ± SEM from n = 4 animals per group. * indicates $p < 0.05$; ** $p < 0.01$; *** $p < 0.005$, and **** $p < 0.0001$.

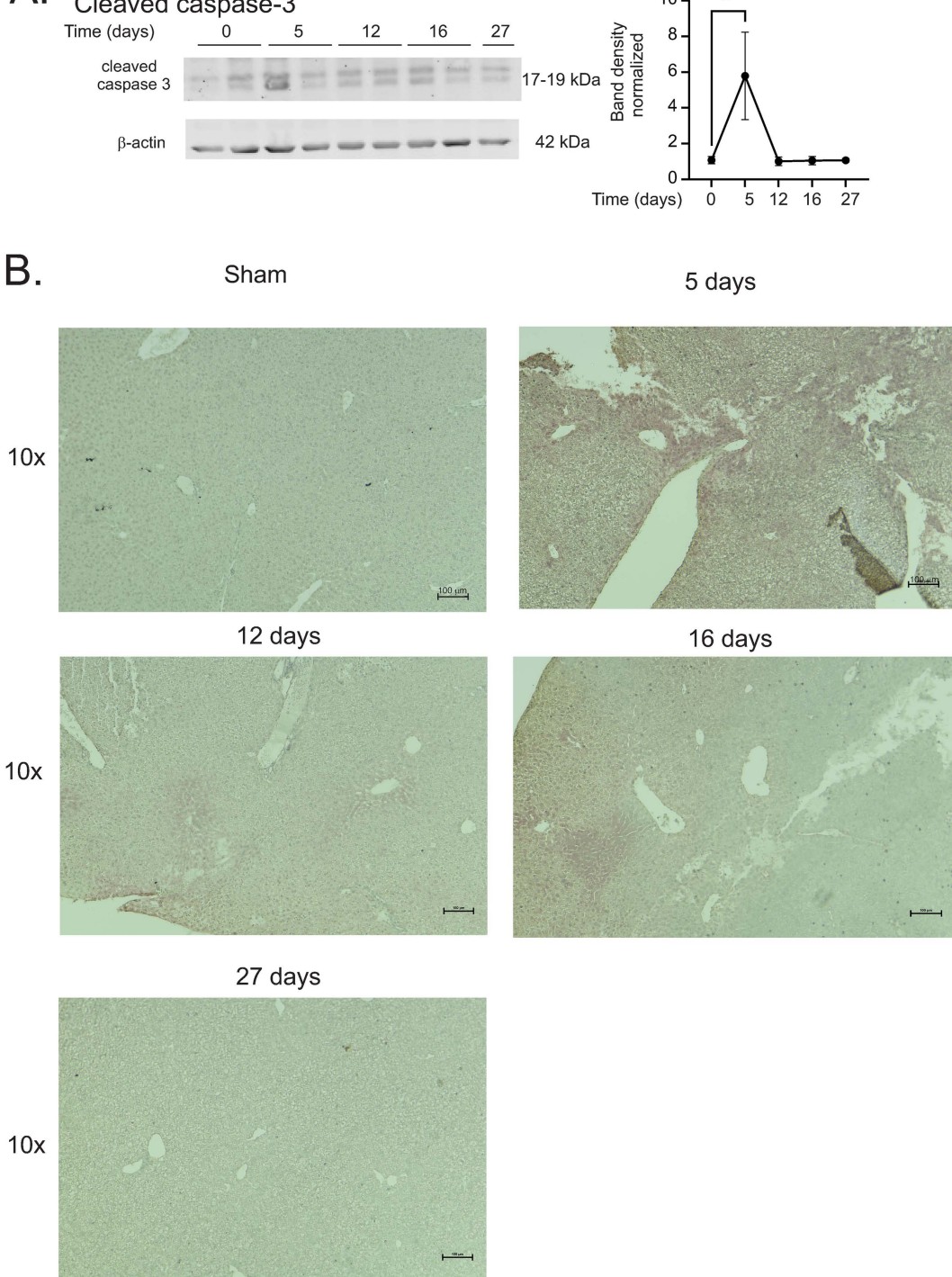

**Fig 4. 7.9 Gy TBI causes early activation of caspase-3 in the liver** C57BL/6 mice were exposed to 7.9 Gy TBI. At the indicated time points, mice were euthanized and liver tissue was obtained for protein analysis. (A) Western blots were performed for active caspase-3; blots were reprobed for β-actin as a loading control. Representative data are shown from n = 2 animals from each time point. Graphs show means ± SEM, n = 4 animals; * indicates $p < 0.05$ from control (0 time point). * indicates $p < 0.05$. (B) Immunohistochemistry of active caspase-3. Representative images are shown, 10 × magnification. Representative images are shown for n = 4 animals per group.

Ferroptosis can be induced by the downregulation of solute carrier family 7 member A11 (*Slc7a11*) and glutathione peroxidase 4 (*Gpx4*), two key enzymes that modulate lipid repair systems [36]. Both *Sc7a11* and *Gpx4* displayed a trend toward biphasic regulation with a slight increase at 5 days post-irradiation, and a subsequent decrease; however, the changes were not significantly different from control levels (Fig. 5A, B). A western blot of GPX-4 protein also showed a trend toward biphasic regulation, but this was also not significantly different from control levels (Fig. 5C). The enzyme Cox-2 (*Ptgs2*), a biomarker but not driver of ferroptosis, was previously shown to be downregulated in ferroptosis [37]. *Ptgs2* also displayed a trend toward downregulation, but the change in its expression did not reach significance (Fig. 5D). These data suggest that early programmed cell death occurs in the liver, as indicated by caspase-3 activation, but markers of ferroptosis are not consistently regulated. The Nrf2 (*Nrf2*) transcription factor is a key response factor that is increased in response to oxidative stress, and is believed to be a pmediator of ferroptosis inhibition [38]. Nrf2 can regulate the expression of *Slc7a11* and *Gpx-4* that modulate ferroptosis [38]. As shown for *Slc7a11* and *Gpx4*, *Nrf2* displayed a

7.9 Gy

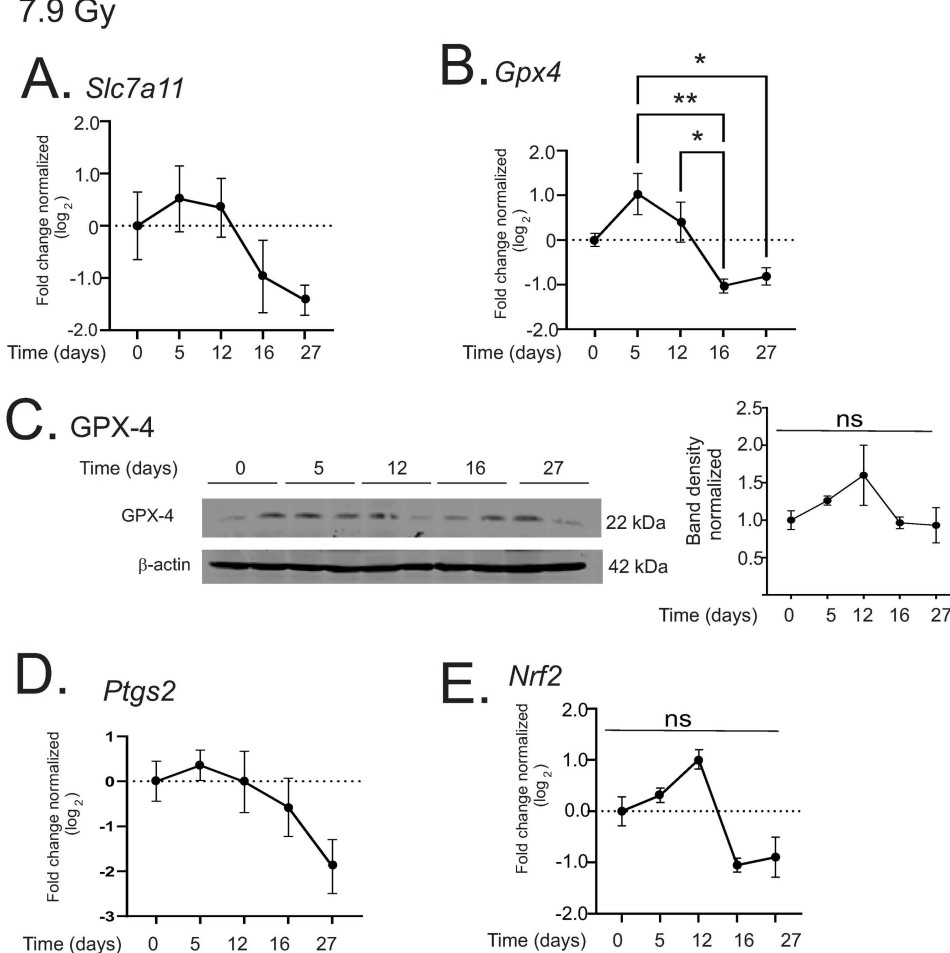

**Fig 5. Regulation of markers of ferroptosis in the liver by 7.9 Gy TBI.** C57BL/6 mice were exposed to 7.9 Gy TBI. At the indicated time points, mice were euthanized and liver tissue was obtained for protein and RNA analysis. (A-B) qPCR was performed to detect solute carrier family 7 A 11 (*Slc7a11*; A), and glutathione peroxidase 4 (*Gpx4*; B). (**C**) Western blot of GPX-4; blot was reprobed for β-actin as a loading control. Representative data are shown from n=2 animals from each group. (D-E) qPCR was performed to detect Cox-2 (*Ptgs2*; C), and Nrf2 (*Nrf2*; D). All graphs show means±SEM, n=4 animals; * indicates $p<0.05$, ** $p<0.01$, *** $p<0.005$, and **** $p<0.0001$.

trend toward biphasic regulation, with a slight increase at 12 days post-irradiation followed by reduced expression; these did not differ significantly from control levels (Fig. 5E).

### Regulation of pro-inflammatory cytokines in the liver following 7.9 Gy TBI

Following TBI, a wide variety of pro-inflammatory cytokines are produced [28]. We investigated the expression of interleukin-1 beta (*Il1b*) and serum amyloid A1 (*Saa1*), two key pro-inflammatory cytokines in the liver following 7.9 Gy TBI. *Il1b* expression was significantly suppressed ~2-fold at all times post-irradiation ($p < 0.05$–$0.001$), but *Saa1* was increased ~3-fold by 17 days post-irradiation ($p < 0.01$, Fig. 6A, B). The increase in *Saa1* corresponds to the time of SAA1 increase in the serum post-irradiation after 7.9 Gy TBI [28].

### Extramedullary hematopoiesis and iron are increased in the liver following exposure to 6.85 Gy TBI

Exposure to 6.85 Gy TBI results in 0–10% mortality in C57BL/6 mice, although red blood cell levels are still significantly reduced at 14 days post-irradiation [2]. Over the time course post-irradiation, erythrophagocytosis was observed, along with mild dilation of sinusoidal spaces (Fig. 7A). No significant indications of cell death were observed histologically. Liver tissue from animals exposed to this 6.85 Gy TBI showed multiple small areas of minimal inflammatory cell aggregates, mostly lymphocytes, especially at 21 days post-irradiation. A mild increase in Prussian blue staining of the liver tissue showed a diffuse, grey coloration of the tissue, especially at 14 and 21 days post-irradiation, although individual cells staining with increased blue color were not detected (Fig. 8A). Iron assays showed that total liver iron levels were increased ~50% compared with control at 14 and 21 days post-irradiation (Fig. 8B $p < 0.05$ and $0.01$, respectively).

### 6.85 Gy TBI results in upregulation of iron-binding proteins in the liver

We again examined changes in expression of iron binding proteins in liver following exposure to the lower dose of TBI. Ferritin showed a steady increase in expression, rising to ~5.5-fold at 21 days post-irradiation ($p < 0.05$, Fig. 9A), corresponding with the time of increased iron levels at this dose of radiation.

The patterns of gene regulation of iron binding/transport proteins differed somewhat from those observed following 7.9 Gy TBI. Transferrin (*Trf*) showed a trend toward biphasic regulation, but this did not reach significance (Fig. 9B). The feline leukemia virus subgroup C receptor 1a (*Flvcr1*) and the *Flvcr2* isoform showed no significant changes in expression (Fig. 9C,

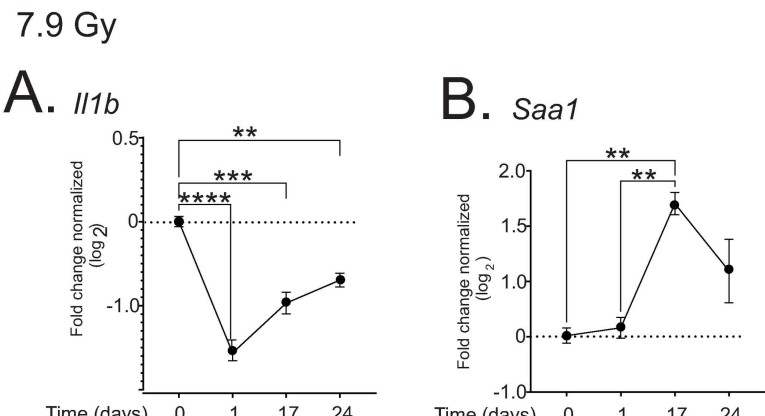

7.9 Gy

**Fig 6. Regulation of inflammatory cytokines in the liver by 7.9 Gy TBI.** C57BL/6 mice were exposed to 7.9 Gy TBI. At the indicated time points, mice were euthanized and liver tissue was obtained for RNA analysis. qPCR was performed to detect IL-1β (*Il1b*; (A)) and SAA-1 (*Saa1*; (B)). Graphs show means ± SEM, n = 4 animals; ** indicates $p < 0.01$, *** $p < 0.005$, and **** $p < 0.0001$.

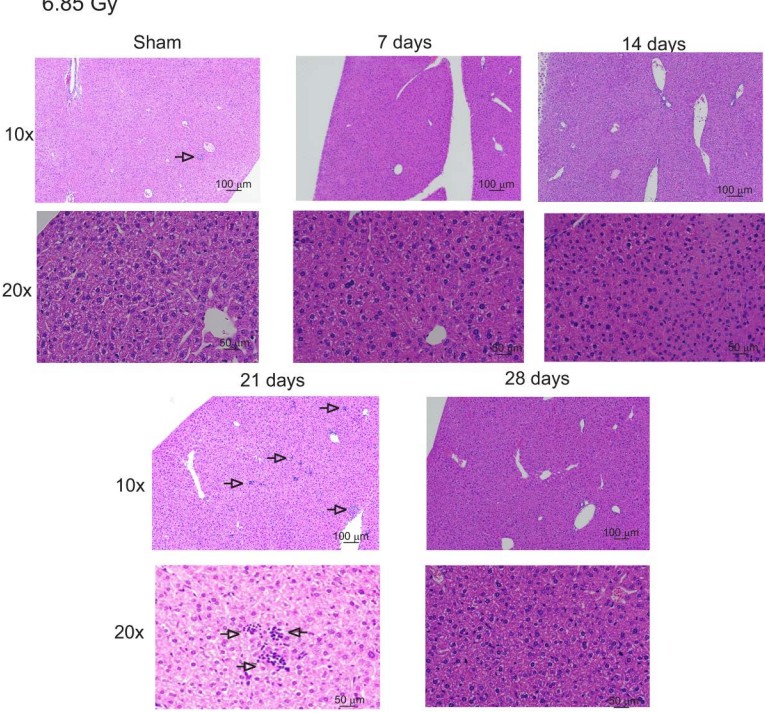

**Fig 7. Histology and iron concentration in mouse livers following 6.85 Gy TBI.** C57BL/6 mice were exposed to 6.85 Gy TBI. At the indicated time points, mice were euthanized, and liver tissue was obtained for histology. Liver sections were stained with H&E to visualize cellular effects of radiation. Representative images from 3 individual animals are shown, with 10× and 40× magnification indicated. Open arrows: leukocyte cell infiltration.

D). In contrast, the integrin M (*Itgam*) displayed significant biphasic regulation, with a 4-fold increase at 7 and 14 days, and ~4-fold decrease from baseline at 21 and 28 days ($p < 0.005$, 0.005, 0.01, and 0.01, respectively; Fig. 9E).

In the group of genes that were suppressed at 7.9 Gy, the transferrin receptor (*Tfrc1*) showed ~2–5-fold *increased* expression at 7–28 days post-irradiation ($p < 0.01$, 0.01, 0.001, and 0.05, respectively; Fig. 9F). In agreement with 7.9 Gy findings, the transferrin receptor 2 (*Tfrc2*) showed a trend toward decreased expression, although this was not significant (Fig. 9G), and the heme carrier protein (*Hcp1*) was decreased 2-fold at 14 days post-irradiation ($p < 0.05$, Fig. 9H).

Finally, in the last group of genes that were not altered in response to 7.9 Gy TBI, lipocalin 2 (*Lcn2*) expression gradually increased, reaching ~4-fold increase at 21 days post-irradiation ($p < 0.01$, Fig. 9I). The divalent metal transporter 1 (*Dmt1*) showed a trend toward decreased expression, especially at 21 days post-irradiation, that did not reach significance (Fig. 9J). In agreement with findings at 7.9 Gy TBI, the *Ho1* gene expression steadily increased post-irradiation, reaching ~2.5-fold at 21 days post-irradiation ($p < 0.001$; Fig. 9K). There was a corresponding increase in HO-1 protein levels to 4-fold, that was significant at 21 days post-irradiation ($p < 0.05$, Fig. 9L). Together these data from 6.85 Gy TBI suggest that increased storage of iron in ferritin occurs in the presence of increased levels of HO-1. Therefore, at 6.85 Gy, the increase in iron in the liver,was associated with increased *Itgam*, *Tfrc1*, and *Lcn2*, suggesting an increase in iron bound to transferrin and iron particle uptake.

## 6.85 Gy TBI induces markers of ferroptosis

Investigation of gene expression changes showed that *Sc7a11* was suppressed ~2-fold, significant at 14 and 21 days post-irradiation ($p < 0.01$ for both; Fig. 10A). *Gpx4* displayed a trend toward suppression at 14–28 days post-irradiation, but this did not reach significance (Fig. 10B). However, GPX-4 protein expression was suppressed at 7–28 days ($p < 0.05$,

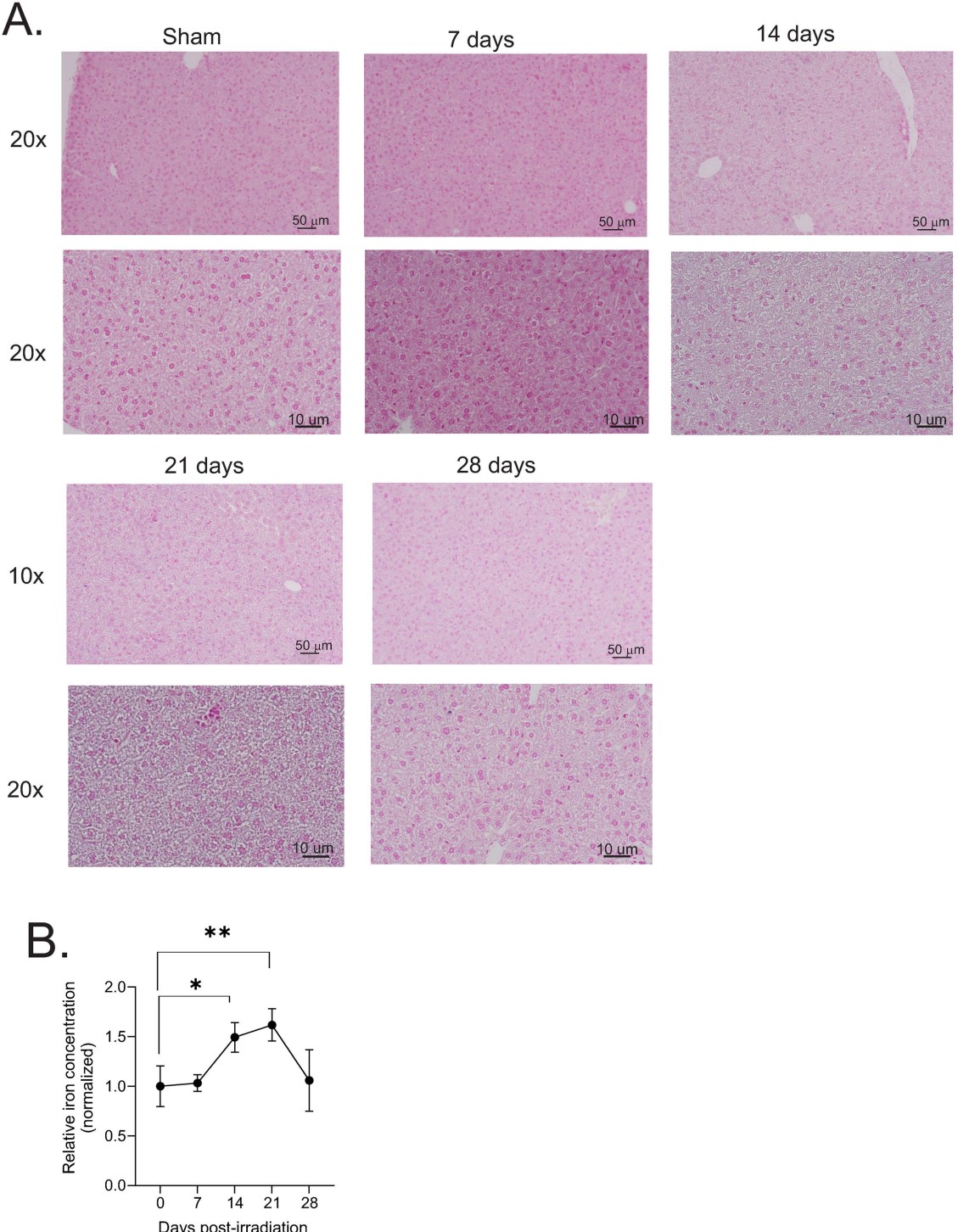

**Fig 8. Iron concentration in mouse livers following 6.85 Gy TBI.** C57BL/6 mice were exposed to 6.85 Gy TBI. At the indicated time points, mice were euthanized, and liver tissue was obtained for histology. **(A)** Prussian blue staining was performed to visualize $Fe^{3+}$ deposition in the tissue. Representative images from 3 individual animals are shown, with 10× and 40× magnification indicated. **(B)** Tissue iron concentration, normalized to protein concentration. Graph shows means±SEM for n=4 animals per group; * indicates $p < 0.05$, ** $p < 0.01$.

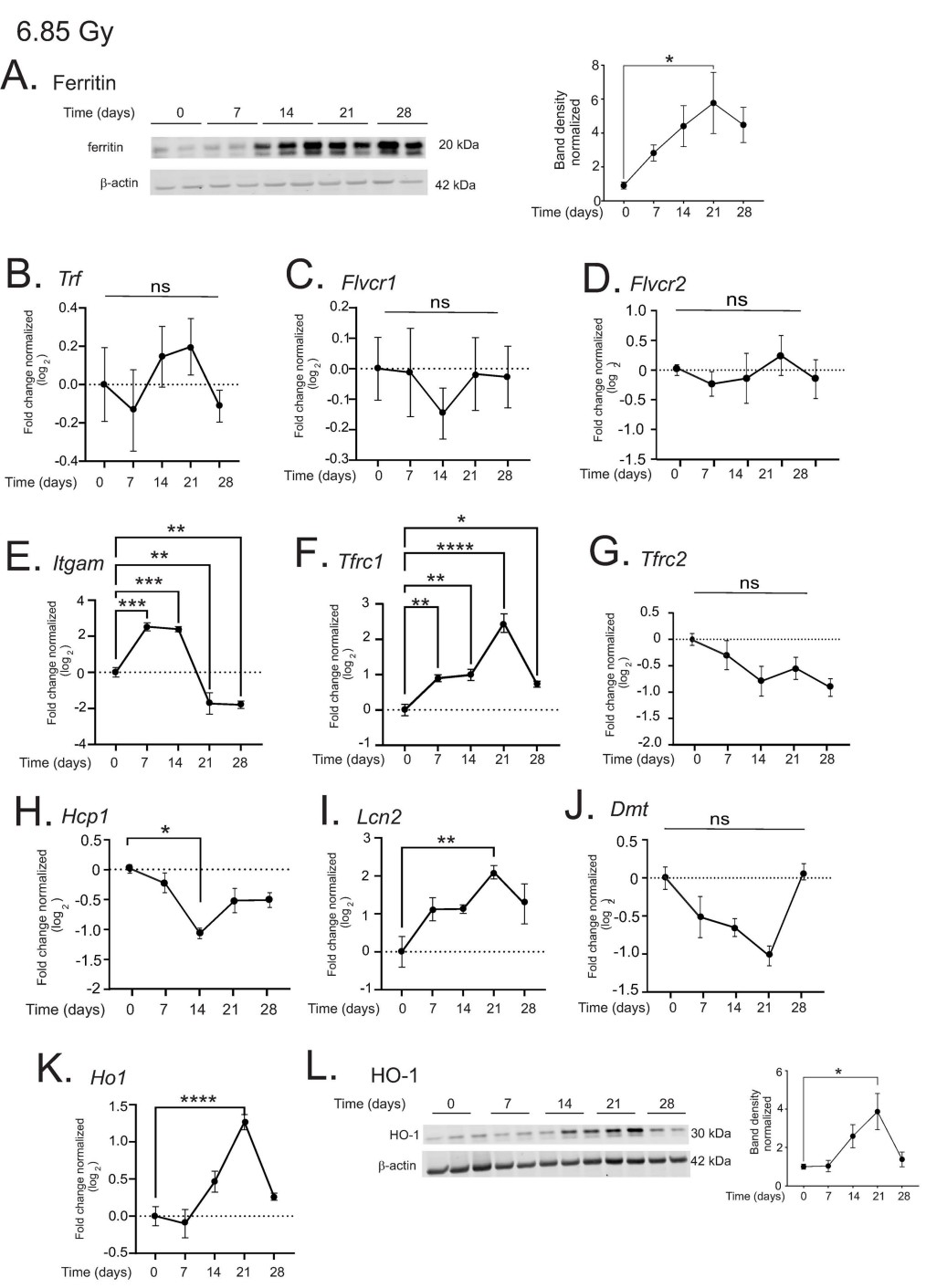

**Fig 9. 6.85 Gy TBI results in increased iron binding proteins in the liver.** C57BL/6 mice were exposed to 6.85 Gy TBI. At the indicated time points, mice were euthanized and liver tissue was obtained for protein and RNA analysis. (A) Western blotting was performed to detect ferritin. A representative blot is shown for n = 4. Graphs show means of band densities ± SEM for n = 4 animals per group, normalized to β-actin protein. B-K. qPCR was performed to determine expression of the following genes: *Trf* (transferrin) (B); *Flvcr1* (feline leukemia virus receptor) (C); *Flvcr2* (D); *Itgam* (integrin alphaM/Mac-1) (E); *Tfrc1* (CD71/transferrin receptor1) (F); *Tfrc2* (G); *Hcp1* (heme carrier protein 1) (H); *Lcn2* (lipocalin-2) (I); *Dmt1* (divalent metal transporter 1) (J); and heme oxygenase-1 (*Ho1*; (K)). (L). Western blotting was performed for HO-1 protein. A representative blot is shown for n = 4. Graphs show means of band densities ± SEM for n = 4 animals per group, normalized to β-actin protein. Graphs shows means ± SEM from n = 4 animals per group. * indicates $p < 0.05$; ** $p < 0.01$; *** $p < 0.005$, and **** $p < 0.0001$.

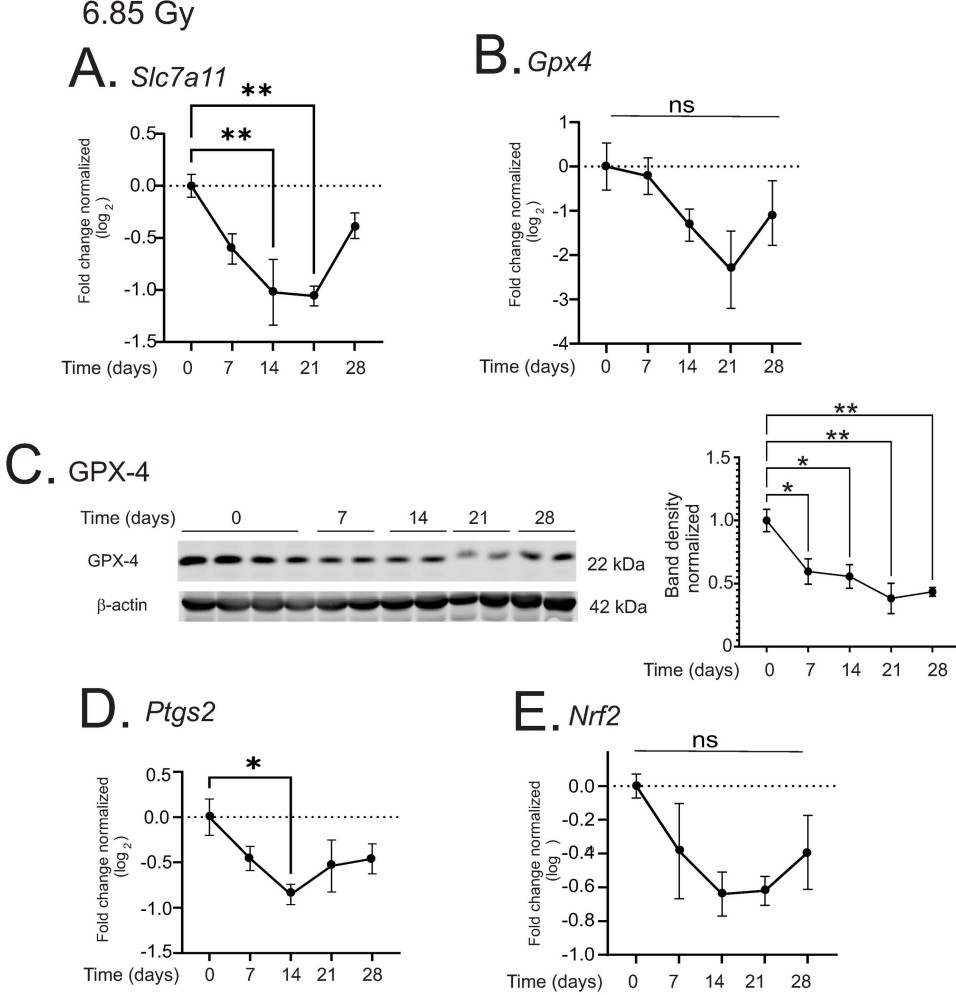

**Fig 10. Regulation of markers of ferroptosis in the liver by 6.85 Gy TBI.** C57BL/6 mice were exposed to 6.85 Gy TBI. At the indicated time points, mice were euthanized and liver tissue was obtained for protein and RNA analysis. (A-B) qPCR was performed to detect solute carrier family 7 A 11 (*Slc7a11*; A), and glutathione peroxidase 4 (*Gpx4*; B). (C) Western blot of GPX-4; blot was reprobed for β-actin as a loading control. Representative data are shown from n = 2 animals from each group. (D-E) qPCR was performed to detect Cox-2 (*Ptgs2*; C), and Nrf2 (*Nrf2*; D). All graphs show means ± SEM, n = 4 animals; * indicates p < 0.05, ** p < 0.01.

0.05, 0.01, and 0.01 respectively; [Fig. 10C]). *Ptgs2* was significant at 14 days post-irradiation ([Fig. 10D]). *Nrf2* displayed a trend toward reduced expression from 7–28 days post-irradiation, although this did not reach significance ([Fig. 10E]). Interestingly, capsase-3 was not significantly regulated over the time course of the experiment, suggesting that there were no significant levels of apoptosis ([Fig. S1]). These data suggest that at 6.85 Gy TBI, a number of markers of ferroptosis are present, but no markers of apoptosis were observed.

## Pro-inflammatory cytokines are upregulated in the liver following 6.85 Gy TBI

We found a greater increase in pro-inflammatory response in the liver following 6.85 Gy compared with 7.9 Gy. *Il1b* expression was significantly increased >8-fold at 21 days post-irradiation (p < 0.001; [Fig 11A]). *Saa1* was also increased ~4–8-fold at 14–21 days post-irradiation (p < 0.005, [Fig. 11B]).

## Discussion

Gamma ionizing radiation exposure induces the rapid production of reactive oxygen and nitrogen species, leading to biomolecular damage that can result in apoptosis and accelerated senescence in affected cells. However, our laboratory and others have shown that a secondary toxic event occurs following partial or total body irradiation due to high levels of iron released from red blood cells (RBC) undergoing hemolysis [12,13,39]. While iron is a necessary element for normal cellular function, high levels of free iron are toxic due to oxidation reactions [18,22]. Iron can be detected in the serum within 1 h after radiation exposures as low as 0.5 Gy [12]. Radiation-induced hemolysis is progressive over time after radiation exposure, with maximal serum iron occurring near time of the RBC nadir [28]. Radiation-induced iron deposition and ferroptosis were demonstrated in mice in the bone, bone marrow, lung, spleen, heart, skin, and intestine following either total and partial body irradiation *in vivo* [1,2,35,39–45]. Recently our laboratory demonstrated the deposition of iron in the tissues of non-human primates following TBI [46]. Studies by several laboratories have shown that iron deposition radiation injuries to multiple organs can be mitigated by iron chelators [35,39–43], suggesting that iron plays a significant role in overall radiation effects. Chelation therapy mitigates radiation damage to the bone marrow, skin, heart, and intestine [34,35,40,47,48]. Here we have demonstrated that TBI, at sublethal or lethal doses of gamma radiation, results in iron deposition in the liver, regulation of iron-binding proteins, and regulation of genes associated with ferroptosis.

Injury of normal liver tissue has been recognized as a limiting factor for the use of radiotherapy for treatment of liver cancer and for treatment of gastrointestinal cancer or other cancers in the anatomical region of the liver [49]. Exposure of the liver to high dose, partial body irradiation, consistent with radiotherapy, results in damage to Kupffer cells, sinusoidal endothelial cells, and hepatic stellate cells. The cellular damage was shown to occur in the presence of oxidative stress (including increased lipid peroxidation and cytotoxic nitric oxide), apoptosis, and the secretion of pro-fibrotic cyokines [49]. Structural changes in the liver following exposure to 8.25 Gy single dose radiation included sinusoidal congestion, hemorrhage, central vein dilation, steatosis, and fibrosis [49]. A recent study to develop a murine model of radiation-induced liver disease, used a single dose of 50 Gy focused irradiation as a model of radiotherapy-induced liver damage [50]. This study showed that alterations in mitochondrial function were associated with fibrotic remodeling at 6 and 20 weeks post-irradiation. An imbalance of reduced and oxidized glutathione was present, especially at days 1–6 post-irradiation. This imbalance could be due to radiation exposure, but the potential contribution of iron deposition was not examined. Studies have demonstrated that iron overload can significantly alter the glutathione oxidation/reduction ratios, lipid peroxidation,

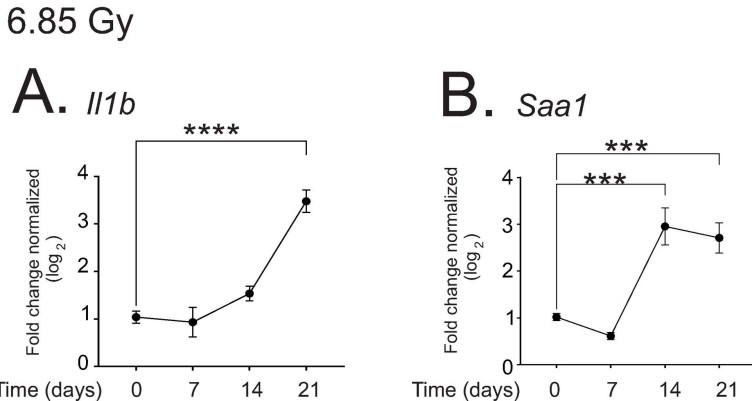

**Fig 11. Regulation of fibrotic cytokines in the liver by 6.85 Gy TBI.** C57BL/6 mice were exposed to 6.85 Gy TBI. At the indicated time points, mice were euthanized and liver tissue was obtained for RNA analysis. qPCR was performed to detect IL-1β (*Il1b*; (**A**) and SAA-1 (*Saa1*; (**B**) Graphs show means ± SEM, n = 4 animals; *** indicates p < 0.005, and **** p < 0.0001.

and fibrotic remodeling [51–53], indicating that there is an overlap between biological processes shown to be activated by radiation and by iron overload.

We compared histological, protein, and gene expression changes in the liver following sublethal (6.85 Gy) and lethal (7.9 Gy) TBI exposures in mice. At 7.9 Gy TBI, we observed hydropic degeneration at 5 days post-irradiation, concurrent with increased apoptosis, followed by micro-vesicular steatosis at 16–27 days. Extramedullary hematopoiesis was present at 12–27 days, suggesting that the liver tissue was able to support hematopoietic progenitors despite the evidence of tissue damage. Expression of the pro-inflammatory cytokine, SAA1, a factor associated with radiation-induced acute inflammatory response, was increased at 17 days post-irradiation, concurrent with steatosis. Our previous studies showed that this is also the time period of radiation-induced acute inflammation in mice as well as other species following TBI [28,54]. At 6.85 Gy, hydropic degeneration, micro-vesicular steatosis, and apoptosis were absent in general. However, extramedullary hematopoiesis was readily apparent, suggesting that the tissue was capable of supporting hematopoietic progenitors. Interestingly, both IL-1β and SAA were produced at ~14–21 days post-irradiation.

We observed iron deposition in the liver at both 7.9 and 6.85 Gy TBI in mice that was maximal at 14–21 days. This time point of maximal accumulation of iron in the liver is after the maximal time points of accumulation in the bone marrow and spleen, suggesting that iron released from these other two organs may be absorbed at the delayed time points by the liver [1,2]. In the liver, both doses of radiation induced the expression of ferritin, the primary iron storage protein in most cell types, but changes in the other iron binding/iron transport proteins differed between the two doses. At 7.9 Gy, we observed the upregulation of transferrin (*Trf*), the primary protein for transport of iron in the serum, but we observed downregulation of the genes for the transferrin receptors as well as for import of heme or iron particles. This suggests that the primary mechanism of iron intake into liver cells may be receptor-independent, and may possibly occur via erythrophagocytosis. At 6.85 Gy we observed increased expression of transferrin receptor 1, lipocalin-2 (Lcn2), and integrin alpha M (*Itgam*), suggesting that receptor-mediated iron uptake was active following 6.85 Gy exposure. Erythrophagocytosis may additionally contribute to the uptake of iron, since damage to erythrocytes would occur at 6.85 Gy [12]. The upregulation of HO-1 at both doses of radiation suggests that iron recycling mechanisms may occur at both 7.9 and 6.85 Gy TBI [55]; further investigation is necessary to determine the contribution of RBC iron to total liver iron after TBI.

As stated above, ferroptosis has been identified as an important secondary effect of TBI [1,2,34,35,39–45]. Although 7.9 Gy TBI resulted in a reduction of *Gpx4* gene expression, we observed only a trend in downregulation of *Slc7a11* and *Ptgs2*. This suggests that ferroptosis is unlikely. Interestingly, we did observe upregulation of active Caspase-3 following 7.9 Gy TBI. In contrast, 6.85 Gy induced significant suppression of GPX-4 protein as well as *Slc7a11* and *Ptgs2* gene expression, indicating a robust regulation of elements that regulate ferroptosis. Caspase-3 activation was observed at the higher dose of radiation. The mechanism(s) for the interaction of ferroptosis and programmed cell death are not completely understood. Ferroptosis can occur in the absence or presence of activated caspases, including caspase-3, depending upon cross-talk with other cell death pathways [37]. Further investigation, using iron chelators or ferroptosis inhibitors together with TBI, would help to elucidate the role of iron versus radiation alone in the induction of cell death in the liver.

Nuclear weapon proliferation and the expansion of nuclear energy and radiation industrial use are an increasing concern due to the risk of accidental or conflict-related radiation exposures. Radiation countermeasures approved by the U.S. Food and Drug Administration (FDA) currently include agents that address acute hematopoietic radiation injury, with a focus on restoring neutrophil and platelet functions [56], but there is an urgent need to develop countermeasures that address other types of acute and delayed tissue injuries [57,58]. This requires expanding our current understanding of the mechanisms of radiation-induced damage to normal tissues for the development of novel radiation countermeasures. Therefore, the development of agents to mitigate secondary iron toxicity may provide an important radiation countermeasure therapy.

## Methods

### Chemicals

Reagents were obtained from Millipore Sigma (St. Louis, MO, USA) except where indicated.

### Animals, captopril treatment, irradiation and tissue collection

The protocol for this murine study was approved by the Institutional Animal Care and Use Committee for USUHS. All animal experiments were performed in strict accordance with the National Institutes of Health Guide for the Care and Use of Laboratory Animals, and the authors have complied with the ARRIVE guidelines. Female C57BL/6 mice were purchased from Jackson Laboratories (Bar Harbor, ME, USA). Mice were kept in a barrier facility for animals accredited by the Association for Assessment and Accreditation of Laboratory Animal Care International. Mice were housed in groups of four in shoebox style cages with individual ventilation, containing standard rodent Beta Chip bedding (Northeastern Products, Warrensburg, NY, USA). Animal rooms were maintained at $21 \pm 2°C$, $50\% \pm 10\%$ humidity, and 12 h light/dark cycle with freely available rodent ration (Harlan Teklad Rodent Diet 8604, Frederick, MD, USA). Adult mice, 12–14 weeks of age (18–22 g body weight), were placed in Lucite jigs and exposed to TBI in a bilateral gamma radiation field in the Armed Forces Radiobiology Research Institute (AFRRI, Bethesda, MD, USA) high level $^{60}$Co facility during morning time periods (7:00–10:00 am) as previously described [2]. The midline tissue dose to the mice was 6.85, 7.85 or 7.90 Gy at a dose rate of 0.6 Gy/min, a sub-lethal radiation exposure for this strain of mice. The alanine/electron spin resonance (ESR) dosimetry system was used to measure dose rates (to water) in the cores of acrylic mouse phantoms. Animals were randomized to treatment groups for each experiment to remove weight difference effects, using four to five animals per group, in accordance with power calculations for these experiments [2]. We determined that sample size of 5 animals per group has 80% power to detect a difference of 2 standard deviations between groups at a given time point based on post-hoc tests for multiple comparisons and a 5% two-sided familywise error rate. If the sample size was as low as 4 animals per group due to mortality, the study had 80% power to detect differences of 2.3 standard deviations between groups. Experimental groups were sham irradiated + vehicle, irradiated + vehicle, with the vehicle being oral administration of 200 μl water on days 2–16 post-irradiation. Sham irradiated control groups were placed in Lucite jigs but without exposure to radiation; the sham irradiated animals were used to determine basal or control levels in experiments. When possible, control groups were used for multiple studies. Following irradiation, animals were evaluated at least twice daily, and three times daily from days 12–27 post-irradiation. Humane endpoints were used for this study, and not death. No time was allowed to elapse from determination that a humane endpoint was reached and euthanasia. The following criteria to evaluate animals twice daily to determine whether euthanasia was required: body weight, behavior, provoked behavior, appearance, and breathing rates. Animals exhibiting loss of more than 20% body weight, lethargy or ataxia, hunched appearance, emaciation or dehydration, or rabid breathing were considered to have reached the humane endpoint and were euthanized. Tissue from animals requiring early euthanasia were not included as the time points would not match the planned time points of the experiment. The experimental endpoints for all surviving animals were 5, 12, 16 and 27 days post-irradiation for 7.9 Gy, and 7, 14, 21, and 28 days post-irradiation for 6.85 Gy. Euthanasia was performed by intraperitoneal injection of 0.1–0.2 ml Fatal Plus (Vortech Pharmaceuticals, Dearborn, MI, USA; 39–78 mg pentobarbital) per animal in accordance with current American Veterinary Medical Association Guidelines for Euthanasia. Euthanasia was ensured prior to collection of tissues. Two independent experiments were conducted for the 7.9 and 6.85 Gy studies.

### Histology and immunohistochemistry (IHC)

Livers were surgically removed from euthanized animals and fixed in 10% neutral buffered formalin overnight. Tissues were processed and embedded in paraffin using standard methods and stained with hematoxylin and eosin (H&E) and Prussian blue staining (Histoserv, Inc., Germantown, MD, USA). Stained slides were evaluated by a pathologist who was blinded to the identity of the treatment groups. IHC for activated caspase-3 was performed on freshly cut sections. Sections were deparaffinized in

xylene, 10 min at room temperature, followed by incubation in 100%, 95%, 70%, and 40% ethanol, each 3 min at room temperature, to rehydrate. Sections were unmasked by boiling for 10 min in citrate-based antigen retrieval buffer, according to the manufacturer's instructions (VectorLabs, Newark, CA, USA; #H-3300). Slides were blocked in 4% normal goat serum (Jackson Immunoresearch, West Grove, PA, USA) in phosphate buffered saline (PBS), 1 h at room temperature. Slides were incubated with anti-active caspase-3 antibody, diluted 1:250 in 2% goat serum in PBS at 4°C overnight (Cell Signaling Technology, Danvers, MA; #9661). Slides were washed 3× in PBS for 9 min at room temperature followed by incubation with goat anti-rabbit antibody diluted 1:200 in 2% goat serum in PBS (Jackson Immunoresearch). Slides were washed as before. Detection was performed using the Vectastain ABC-AP kit followed by development using the Vector red alkaline phosphatase substrate kit (VectorLabs), both according to the manufacturer. Slides were dehydrated by a reversal of the hydration process. Images were obtained at 10–40× magnification, using a Nikon Eclipse Ti microscope (Minato City, Tokyo, Japan).

## Western blotting

At the indicated time, cells were lysed in RIPA buffer (1% NP-40, 0.1% SDS, 0.1% Na-deoxycholate, 10% glycerol, 0.137 M NaCl, 20 mM Tris pH [8.0]) (Thermo Fisher Scientific, Waltham, MA, USA), protease(#A32953, Thermo Fisher) and phosphatase (#A32957, Thermo Fisher Scientific) inhibitor cocktails for 20 min at 4°C, vortexed, rotated at 4° C for 20 min, then centrifuged at 6,000 × rcf for 7 min. Protein concentrations were determined using the BCA protein assay (MilliporeSigma, Burlington, MA, USA). Clarified lysates were boiled in SDS sample buffer mixture containing 10 × DTT and 4 × Laemmli Sample Buffer (Bio-Rad, Hercules, CA, USA; #161–0747) for 5 min prior to resolution by sodium dodecyl sulfate-polyacrylamide gel electrophoresis. 12% Polyacrylamide gel: 3.15 mL dH$_2$O, 3.6 mL 30% acrylamide/bis solution (Bio-Rad #161–0158), 2.25 mL 1.5 Tris-HCl resolving gel buffer (Bio-Rad, #161–0798), 90 µL 10% SDS solution (Bio-Rad #1610416), 45 µL 10% ammonium persulfate (Bio-Rad, #161–0700), 3 µL TEMED (Bio-Rad, #161–0800). Proteins were transferred to nitrocellulose membranes with 2 µm pore size (MilliporeSigma). Nitrocellulose membranes were exposed to Everyblot blocking agent (Bio-Rad, #12010020) for 1.5 hours. Proteins were identified using primary antibodies diluted in 5% BSA in TBS 0.5% tween 20 buffer. Western blot protein bands were detected and quantified using the Odyssey system (LI-COR, Lincoln, NE). Proteins were detected by blotting using anti-ferritin (Santa Cruz Biotechnology, Santa Cruz, CA, USA; #sc-256; 1:1000), anti-ferroportin (Novus Biologicals, Centennial, CO, USA; #NBP1–21502SS; 1:1000) anti-β-actin (Millipore Sigma #AC-15; 1:5,000), anti-heme oxygenase 1 (Cell Signaling, Danvers, MA, USA; #43966S; 1:1000), anti-GPX4 (Cell Signaling, #59735S; 1:250), anti-cleaved caspase-3 (Cell Signaling #9662; 1:1000); and anti-Cox-2 (Cell Signaling, #4842; 1:1000) primary antibodies. Antibodies were diluted in the buffers recommended by the manufacturers and incubated 4°C for 1–5 days with gentle rocking. Anti-mouse and anti-rabbit secondary antibodies conjugated to IRDye680 or IRDye800 (LI-COR; 1:10,000) were used to probe primary antibodies, according to the manufacturer's instructions. Proteins were normalized to β-actin which was used to probe the same gel. Gel images were prepared using Odyssey System software and Corel Draw X7 Graphics software (Corel Corporation, Ottawa, Ontario, Canada), in accordance with the Author Guidelines. Full gels are provided in the supporting materials.

## Iron quantification

Levels of total iron (II + III) were determined in liver homogenates using the Iron Colorimetric Assay Kit (Biovision Inc., Milpitas, CA, USA) according to the manufacturer's instructions. Iron concentrations were normalized to protein concentrations, determined by BCA protein assay (MilliporeSigma).

## RNA isolation and quantitative PCR analysis

Liver tissues were immediately placed in RNAlater following euthanasia, and were stored according to the manufacturer's instructions (Qiagen, Germantown, MD, USA). At the time of RNA isolation, tissues were homogenized with an Ultra Turrax homogenizer (Jahnke & Kunkel, Staufen, Germany). RNA was isolated using the Qiagen RNeasy Mini Kit with

on-column DNase digestion (Qiagen, Valencia, CA, USA) according to manufacturer's protocol. RNA concentrations were determined spectroscopically using a Nano-drop (ND-1000 Spectrophotometer, Nano-Drop, Wilmington, DE, USA). 1.0 μg RNA was reverse transcribed using iScript cDNA synthesis kit (Bio-Rad) in an iCycler containing an IQ5 optical system (Bio-Rad). RT-PCR reactions were performed with technical duplicates using iTaqTM Universal SYBR Green Supermix (Bio-Rad) in a CFX96 Touch Real-Time PCR Detection System (Bio-Rad), or with PowerSYBR Green PCR Master Mix (ThermoFisher) using the QuantStudio7 Pro Real-Time PCR detection system (ThermoFisher) [2]. Each reaction contained 1×SYBR Green (Bio-Rad), 10 ng cDNA, and 12 μM primers. Primers for qRT-PCR were designed using NCBI/ Primer–BLAST and purchased from Integrated DNA Technologies (Coralville, IA, USA) (Table 1). Genes were normalized to *Gapdh*. Relative gene expression to the reference genes was calculated using the ΔΔCq method using CFX Maestro software, 2.0 (Bio-Rad) [41,42].

## Statistical analysis

Statistical analysis was performed using GraphPad Prism V6 (GraphPad Prism Software, Inc., San Diego, CA, USA). Results are represented as means±SEM. P values of < 0.05 were considered significant. One-way ANOVA with either Tukey's or Dunnett's post-hoc test was used for multiple comparisons.

**Table 1. Murine gene primers for qPCR.**

| Gene | Forward Primer | Reverse Primer |
| --- | --- | --- |
| DMT1 | 5'-TTGCTCCTGGGATATGGAGT-3' | 5'- TGCTGTAGGCAGGGTTGATG-3' |
| Flvcr1 | 5'-GGCACAATATAAACACCGGGC-3' | 5'-TCCGACTGTATAGACACCATGAC-3' |
| Flvcr2 | 5'- GCCTGGGAGCCTGGGTAAAG-3' | 5'-ATGCTGATGTGGTAGGCAAGC -3' |
| Fth1 | 5'-AGTGCGCCAGAACTACCAC-3' | 5'-AGCCACATCATCTCGGTCAA-3' |
| Gapdh | 5'-ATGTGTCCGTTGTGGACTTG-3' | 5'-GGTCCTCAGTGTAGCCCAAG-3' |
| Gpx4 | 5'-CGCCAAAGTCCTAGGAAACG-3' | 5'-AAGGTTCAGGAATGGGCTCC-3' |
| HCP1 | 5'-AGGCTTCTGCAATTCTGCCT -3' | 5'-CCACAGCAGAGAACAGAGCA -3' |
| Ho1 | 5'-GTTTGAGGAGCTGCAGGTGA -3' | 5'-TGCCAACAGGAAGCTGAGAG -3' |
| IL1b | 5'-GTG TCT TTC CCG TGG ACC TT-3' | 5'-TCG TTG CTT GGT TCT CCT TG-3' |
| Itgam | 5'-AGAACACCAAGGACCGTCTG-3' | 5'-AATCCAAAGACCTGGGTGCG-3' |
| Lcn2 | 5'-GGACTACAACCAGTTCGCCA-3' | 5'-CAAAGCGGGTGAAACGTTCC-3' |
| Nrf2 | 5'-CCTCACCTCTGCTGCAAGTA -3' | 5'-AACTTGTACCGCCTCGTCTG -3' |
| Ptgs2 | 5'-CGGADAGAGTTCATCCCTGAC -3' | 5'- CAGGGATGTGAGGGTAG-3' |
| Rpl13 | 5'-CTTTTCCCAGACGAGGATATTCC-3' | 5'-CCAGCCGTTTAGGCACTCT-3' |
| Saa1 | 5'-TCAGACAAATACTTCCATGCT-3' | 5'-AAAGGCCTCTCTTCCATCACT-3' |
| Slc11a11 | 5'-TCCGAGGAGCAAGAGGAGTAA-3' | 5'-TCCCCTTTGCTATCACCGAC-3' |
| Slc40a1 | 5'-TTCCTCCTCTACCTTGGCCA-3' | 5'-CTGCCACCACCAGTCCATAG-3' |
| Trf1 | 5'-AAGTGCATCAGCTTCCGTGA-3' | 5'-AGACCACACTGGCCTTGATG-3' |
| Tfrc1 | 5'-GCTCGTGGAGACTACTTCCG-3' | 5'-AGAGAGGGCATTTGCGACTC-3' |
| Tfrc2 | 5'-CTGGCTTCCCGTCCTTCAAT -3' | 5'-CAGCCGATAAGGAGAGCCTG -3' |
| Ywhaz | 5'-CTTTCTGGTTGCGAAGCATT-3' | 5'-TTGAGCAGAAGACGGAAGGT-3' |

Murine sequences for DMT1 (divalent metal transporter 1), *Flvcr1* and *2* (feline leukemia virus receptor 1 and 2; heme iron receptor), *Fth1* (ferritin heavy chain 1), *Gapdh* (glyceraldehyde-3-phosphate dehydrogenase), *Gpx4* (glutathione peroxidase 4), *HAMP* (heme carrier protein 1), *Il1b* (interleukin 1 beta), *Il6* (interleukin-6), *Itgam* (integrin subunit alpha M), *Lcn2* (lipocalin-2; also known as neutrophil gelatinase associated lipocain or siderocalin), *Nos2* (inducible nitric oxide synthase), *Slc7a11* (solute carrier 7 member 11), *Scl40a1* (solute carrier 20 member 1, ferroportin),*Tfrc1* and 2 (transferrin receptor-1 and -2), *Trf1* (transferrin-1).

## Supporting information

**S1 Fig. 6.85 Gy TBI does not lead to caspase-3 activation in the liver.** C57BL/6 mice were exposed to 6.85 Gy TBI. At the indicated time points, mice were euthanized and liver tissue was obtained for protein analysis. Western blots were performed for total and activated caspase-3; blots were reprobed for β-actin as a loading control. Representative data are shown from n = 4 control (sham-irradiated) animals and n = 2 animals from each time point post-irradiation.
(TIF)

**S2 Raw Images. Unprocessed western blot images.**
(PDF)

**S3 Raw data file.** Raw data for all qPCR, iron assays, and western blot data.
(DOCX)

## Acknowledgments

Some of the authors are employees of the US Government. This work was prepared as a part of their official duties and therefore is in the public domain and does not possess copyright protection. Public domain information may be freely distributed and copied. However, as a courtesy it is requested that the Uniformed Services University and the authors be given appropriate acknowledgement. The opinions and assertions in this article are those of the authors and do not necessarily reflect the official policy or position of the Uniformed Services University of the Health Sciences, the Armed Forces Radiobiology Research Institute, Department of the Navy, Department of Defense or the U.S. Federal Government.

## Author contributions

**Conceptualization:** Sang-Ho Lee, Regina M. Day.

**Data curation:** Dmitry T. Bradfield, John E. Slaven, W. Bradley Rittase, Milan Rusnak, Aviva J. Symes, Grace V. Brehm, Jeannie M. Muir, Sang-Ho Lee, Joseph A. Anderson, Regina M. Day.

**Formal analysis:** Dmitry T. Bradfield, John E. Slaven, W. Bradley Rittase, Milan Rusnak, Aviva J. Symes, Grace V. Brehm, Jeannie M. Muir, Sang-Ho Lee, Joseph A. Anderson, Regina M. Day.

**Funding acquisition:** Aviva J. Symes, Regina M. Day.

**Investigation:** Dmitry T. Bradfield, W. Bradley Rittase, Milan Rusnak, Jeannie M. Muir, Sang-Ho Lee, Joseph A. Anderson, Regina M. Day.

**Methodology:** Dmitry T. Bradfield, John E. Slaven, W. Bradley Rittase, Jeannie M. Muir, Regina M. Day.

**Project administration:** Aviva J. Symes, Regina M. Day.

**Supervision:** Regina M. Day.

**Validation:** Joseph A. Anderson, Regina M. Day.

**Writing – original draft:** Regina M. Day.

**Writing – review & editing:** Dmitry T. Bradfield, John E. Slaven, W. Bradley Rittase, Milan Rusnak, Aviva J. Symes, Grace V. Brehm, Jeannie M. Muir, Sang-Ho Lee, Joseph A. Anderson.

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
