## [Decision Letter · Decision Letter 0]

1 Oct 2024

PONE-D-24-37950Cell Death and Iron Deposition in the Liver in two Murine Models of Acute Radiation SyndromePLOS ONE

Dear Dr. Day,

Thank you for submitting your manuscript to PLOS ONE. After careful consideration, we feel that it has merit but does not fully meet PLOS ONE’s publication criteria as it currently stands. Therefore, we invite you to submit a revised version of the manuscript that addresses the points raised during the review process.

Please submit your revised manuscript by Nov 15 2024 11:59PM. If you will need more time than this to complete your revisions, please reply to this message or contact the journal office at plosone@plos.org . Please include the following items when submitting your revised manuscript:

We look forward to receiving your revised manuscript.

Kind regards,

Hiroyasu Nakano, M.D., Ph.D.

Academic Editor

PLOS ONE

Journal Requirements:

2. Thank you for your submission to PLOS ONE. We note that your study design may include death of a regulated animal as a likely outcome or planned experimental endpoint. At this time, we request that you please report additional details in your Methods section regarding animal care and use for the survival study, as per our editorial guidelines (http://journals.plos.org/plosone/s/submission-guidelines#loc-humane-endpoints). 

For easy reference, we have attached a checklist that may be relevant for your submission. Please complete all items on the checklist at the following link:   

http://journals.plos.org/plosone/s/file?id=bb1d/plos-one-humane-endpoints-checklist.docx         

Please upload the completed checklist as file type “Other” when resubmitting your manuscript. This document is for internal journal use only and will not be published if your article is accepted. We very much appreciate your attention to these requests and support of improved reporting standards in PLOS ONE submissions.

4. We note that your Data Availability Statement is currently as follows: 

“All relevant data are within the manuscript and its Supporting Information files.”

7. Your ethics statement should only appear in the Methods section of your manuscript. If your ethics statement is written in any section besides the Methods, please move it to the Methods section and delete it from any other section. Please ensure that your ethics statement is included in your manuscript, as the ethics statement entered into the online submission form will not be published alongside your manuscript. 

**Additional Editor Comments:**

While both reviewers find the study potentially interesting, they have expressed significant concerns regarding the lack of clear evidence demonstrating apoptotic cells in the liver. Therefore, the authors should address this issue.

Reviewers' comments:

Reviewer's Responses to Questions

**Comments to the Author**

1. Is the manuscript technically sound, and do the data support the conclusions?

Reviewer #1: Partly

Reviewer #2: Partly

2. Has the statistical analysis been performed appropriately and rigorously? 

Reviewer #1: Yes

Reviewer #2: Yes

3. Have the authors made all data underlying the findings in their manuscript fully available?

Reviewer #1: Yes

Reviewer #2: Yes

4. Is the manuscript presented in an intelligible fashion and written in standard English?

Reviewer #1: Yes

Reviewer #2: Yes

5. Review Comments to the Author

Reviewer #1: In this manuscript, the authors investigated the relationship between tissue injury and iron-related proteins in mouse liver with high-dose total body irradiation (TBI). They already demonstrated 7.9 Gy TBI induced tissue injury but not 6.85 Gy. TBI was reported to induce the hemolysis leading to the release of iron, and they confirmed the effects of released iron on TBI-induced tissue injury. At first, they might intend to connect ferroptosis with tissue injury. However non-toxic 6.85 Gy treatment showed the clear changes related to ferroptosis, but toxic 7.9 Gy treatment did not. Although there is some difference related to iron transport between two conditions, ferroptosis was not the major reason for tissue injury. In contrast, they described “7.9 Gy induced liver caspase-3 activation consistent with apoptosis. In contrast, 6.85 Gy induced markers of ferroptosis but not of apoptosis” in abstract and “Caspase-3 activation was not observed at this lower dose of radiation (data not shown)” in discussion. These parts imply the importance of apoptosis or caspase activation in liver tissue injury, but the authors did not show the results examining caspase-3 activity in 6.85 Gy-treated sample. Although the results were confusing and did not support their hypothesis, they should show essential data related to apoptosis as well as ferroptosis. The experimental data and discussions are sufficient for the publication in PlosOne with additional data related to apoptosis.

Reviewer #2: In this study, Day and colleagues analyzed the kinetics of proteins and genes in the liver of mice subjected to total body irradiation (TBI). They found that TBI may induce iron accumulation, as indicated by increased levels of ferritin protein. Additionally, upregulation of HO-1 may be associated with increased oxidative stress in the liver. This study provides valuable histopathological data that could help evaluate the degree of post-irradiation liver injury not only in mice but also in humans. The introduction is well-written and explains the mechanisms underlying TBI-induced iron dysregulation. However, some statements lack adequate scientific support, and cell death was not extensively analyzed in the data provided. Overall, while the value of this work is undeniable, this reviewer recommends that the authors carefully revise the manuscript as outlined below before publication in PLOS ONE.

Major Comments

1. Although the authors state that TBI induced hepatocellular apoptosis, Figure 3 does not show a clear upregulation of activated caspase-3 over the course of the experiment. To support this claim, apoptosis induction should also be confirmed by immunohistochemical analysis of the liver. The number of activated caspase-positive hepatocytes should be quantified for statistical analysis.

2. The title of this paper is somewhat misleading, as markers of ferroptosis were not significantly upregulated in the liver of TBI mice. It is generally recommended that the authors demonstrate inhibition of cell death using ferroptosis inhibitors to substantiate claims of ferroptosis involvement.

Minor Comments

1. The authors state that TBI induces erythrophagocytosis by Kupffer cells (KCs) in the liver (e.g., Figures 1 and 6). To support this claim, macrophages should be specified using CD68 or F4/80 staining.

2. The authors claim that the Prussian Blue-positive area increased post-TBI. However, Prussian Blue-positive areas were not clearly evident in any of the provided images.

6. PLOS authors have the option to publish the peer review history of their article (what does this mean? ). If published, this will include your full peer review and any attached files.

**Do you want your identity to be public for this peer review?** For information about this choice, including consent withdrawal, please see our Privacy Policy .

Reviewer #1: No

Reviewer #2: No

---

## [Author Response · Author response to Decision Letter 1]

9 Apr 2025

Specific Reviewer comments:

Reviewer #1: In this manuscript, the authors investigated the relationship between tissue injury and iron-related proteins in mouse liver with high-dose total body irradiation (TBI). They already demonstrated 7.9 Gy TBI induced tissue injury but not 6.85 Gy. TBI was reported to induce the hemolysis leading to the release of iron, and they confirmed the effects of released iron on TBI-induced tissue injury. At first, they might intend to connect ferroptosis with tissue injury. However non-toxic 6.85 Gy treatment showed the clear changes related to ferroptosis, but toxic 7.9 Gy treatment did not.

Our investigation of ferroptosis indicates that there are markers of ferroptosis present in the liver following 6.85 Gy but not in 7.9 Gy. Conclusive evidence of ferroptosis should include either data showing oxidized lipid modification of proteins or direct lipid oxidation. We were unable to obtain this conclusive evidence with several assays that we conducted. We now state in the Results of the 7.9 Gy study that our data suggest that markers of ferroptosis are not present; we also state in the Results of the 6.85 Gy section that markers of ferroptosis are present. We now state in the Conclusions section that the changes in ferroptosis markers are present following 6.85 Gy exposure, but that additional data would be required to provide proof of ferroptosis.

Although there is some difference related to iron transport between two conditions, ferroptosis was not the major reason for tissue injury. In contrast, they described “7.9 Gy induced liver caspase-3 activation consistent with apoptosis. In contrast, 6.85 Gy induced markers of ferroptosis but not of apoptosis” in abstract and “Caspase-3 activation was not observed at this lower dose of radiation (data not shown)” in discussion. These parts imply the importance of apoptosis or caspase activation in liver tissue injury, but the authors did not show the results examining caspase-3 activity in 6.85 Gy-treated sample.

We examined Caspase-3 activation after 6.85 Gy exposure, and our data was negative. This negative data is now shown in Figure S1.

Reviewer #2: In this study, Day and colleagues analyzed the kinetics of proteins and genes in the liver of mice subjected to total body irradiation (TBI). They found that TBI may induce iron accumulation, as indicated by increased levels of ferritin protein. Additionally, upregulation of HO-1 may be associated with increased oxidative stress in the liver.

This study provides valuable histopathological data that could help evaluate the degree of post-irradiation liver injury not only in mice but also in humans. The introduction is well-written and explains the mechanisms underlying TBI-induced iron dysregulation. However, some statements lack adequate scientific support, and cell death was not extensively analyzed in the data provided. Overall, while the value of this work is undeniable, this reviewer recommends that the authors carefully revise the manuscript as outlined below before publication in PLOS ONE.

Major Comments

1. Although the authors state that TBI induced hepatocellular apoptosis, Figure 3 does not show a clear upregulation of activated caspase-3 over the course of the experiment. To support this claim, apoptosis induction should also be confirmed by immunohistochemical analysis of the liver. The number of activated caspase-positive hepatocytes should be quantified for statistical analysis.

We were only able to show caspase-3 activation transiently at 5 days post-irradiation. In the future, a tighter time course, with more early time points, may be able to provide better proof of activation of markers of apoptosis in the liver. We now provide IHC

2. The title of this paper is somewhat misleading, as markers of ferroptosis were not significantly upregulated in the liver of TBI mice. It is generally recommended that the authors demonstrate inhibition of cell death using ferroptosis inhibitors to substantiate claims of ferroptosis involvement.

A number of studies have demonstrated that the inclusion of ferroptosis inhibitors or iron chelators reduce tissue pathology following total body or partial body irradiation. Unfortunately, we do not have specific funding to support these animal studies at this time. Without a specific peer-reviewed grant to perform this work, we are unable to perform these studies. We hope to obtain funding to pursue this work in the future. We have amended the title to include iron deposition and we have removed cell death and ferroptosis.

Minor Comments

1. The authors state that TBI induces erythrophagocytosis by Kupffer cells (KCs) in the liver (e.g., Figures 1 and 6). To support this claim, macrophages should be specified using CD68 or F4/80 staining.

These cells were identified by a certified veterinary pathologist. We have removed this claim from the manuscript.

2. The authors claim that the Prussian Blue-positive area increased post-TBI. However, Prussian Blue-positive areas were not clearly evident in any of the provided images.

The Prussian Blue staining was diffuse in most cases, and appears as a grey-blue discoloration of the tissues. We provide additional images of the Prussian Blue stained tissues. Additionally, we provide the iron assays of the liver tissue after both levels of radiation exposure, that provide quantification of the amount of iron increased in the tissues.

---

## [Decision Letter · Decision Letter 1]

24 Apr 2025

Cell Death and Iron Deposition in the Liver in two Murine Models of Acute Radiation Syndrome

PONE-D-24-37950R1

Dear Dr. Day,

We’re pleased to inform you that your manuscript has been judged scientifically suitable for publication and will be formally accepted for publication once it meets all outstanding technical requirements.

Kind regards,

Hiroyasu Nakano, M.D., Ph.D.

Academic Editor

PLOS ONE

Additional Editor Comments (optional):

Reviewers' comments:

Reviewer's Responses to Questions

**Comments to the Author**

1. If the authors have adequately addressed your comments raised in a previous round of review and you feel that this manuscript is now acceptable for publication, you may indicate that here to bypass the “Comments to the Author” section, enter your conflict of interest statement in the “Confidential to Editor” section, and submit your "Accept" recommendation.

Reviewer #1: All comments have been addressed

Reviewer #2: (No Response)

2. Is the manuscript technically sound, and do the data support the conclusions?

Reviewer #1: Yes

Reviewer #2: Partly

3. Has the statistical analysis been performed appropriately and rigorously? 

Reviewer #1: Yes

Reviewer #2: I Don't Know

4. Have the authors made all data underlying the findings in their manuscript fully available?

Reviewer #1: Yes

Reviewer #2: Yes

5. Is the manuscript presented in an intelligible fashion and written in standard English?

Reviewer #1: Yes

Reviewer #2: Yes

6. Review Comments to the Author

Reviewer #1: In the revised manuscript, the authors included the data in response to my comment, showing that apoptosis is not involved in 6.85 Gy TBI. This data clearly supports the conclusion that apoptosis is important for highly-toxic 7.9 Gy TBI, but not for low toxic 6.85 Gy TBI. Although the involvement of ferroptosis remains unclear at this stage, this manuscript is now suitable for publication in PlosOne.

Reviewer #2: Thank you for your efforts in revising the manuscript and addressing the reviewer’s comments. I appreciate the improvements you have made, which demonstrate your commitment to refining your work.

Although there are still areas that could have been addressed more thoroughly to further enhance the quality and clarity of the manuscript, I acknowledge the substantial effort put into responding to the review. On this basis, I find the revised manuscript suitable for publication.

7. PLOS authors have the option to publish the peer review history of their article (what does this mean? ). If published, this will include your full peer review and any attached files.

**Do you want your identity to be public for this peer review?** For information about this choice, including consent withdrawal, please see our Privacy Policy .

Reviewer #1: No

Reviewer #2: No

---

## [Editor Report · Acceptance letter]

PONE-D-24-37950R1

PLOS ONE

Dear Dr. Day,

I'm pleased to inform you that your manuscript has been deemed suitable for publication in PLOS ONE. Congratulations! Your manuscript is now being handed over to our production team.

Kind regards,

on behalf of

Professor Hiroyasu Nakano

Academic Editor

PLOS ONE